# Learning Federated Neural Graph Databases for Answering Complex Queries from Distributed Knowledge Graphs

**Qi Hu**                                                      *qhuaf@connect.ust.hk*
*The Hong Kong University of Science and Technology*

**Weifeng Jiang**                                        *weifeng001@e.ntu.edu.sg*
*Nanyang Technological University*

**Haoran Li**                                                *hlibt@connect.ust.hk*
*The Hong Kong University of Science and Technology*

**Zihao Wang**                                          *zwanggc@connect.ust.hk*
*The Hong Kong University of Science and Technology*

**Jiaxin Bai**                                                *jbai@connect.ust.hk*
*The Hong Kong University of Science and Technology*

**Qianren Mao**                                          *maoqr@zgclab.edu.cn*
*Zhongguancun Laboratory*

**Yangqiu Song**                                          *yqsong@cse.ust.hk*
*The Hong Kong University of Science and Technology*

**Lixin Fan**                                              *fanlixin@webank.com*
*WeBank*

**Jianxin Li**                                            *lijx@act.buaa.edu.cn*
*Beihang University*

**Reviewed on OpenReview:** *https://openreview.net/forum?id=3K1LRetR6Y*

## Abstract

The increasing demand for deep learning-based foundation models has highlighted the importance of efficient data retrieval mechanisms. Neural graph databases (NGDBs) offer a compelling solution, leveraging neural spaces to store and query graph-structured data, thereby enabling LLMs to access precise and contextually relevant information. However, current NGDBs are constrained to single-graph operation, limiting their capacity to reason across multiple, distributed graphs. Furthermore, the lack of support for multi-source graph data in existing NGDBs hinders their ability to capture the complexity and diversity of real-world data. In many applications, data is distributed across multiple sources, and the ability to reason across these sources is crucial for making informed decisions. This limitation is particularly problematic when dealing with sensitive graph data, as directly sharing and aggregating such data poses significant privacy risks. As a result, many applications that rely on NGDBs are forced to choose between compromising data privacy or sacrificing the ability to reason across multiple graphs. To address these limitations, we propose to learn Federated Neural Graph DataBase (FedNGDB), a pioneering systematic framework that empowers privacy-preserving reasoning over multi-source graph data. FedNGDB leverages federated learning to collaboratively learn graph representations across multiple sources, enriching relationships between entities, and improving the overall quality of graph data. Unlike existing methods, FedNGDB can handle complex graph structures and re-

lationships, making it suitable for various downstream tasks. We evaluate FedNGDBs on three real-world datasets, demonstrating its effectiveness in retrieving relevant information from multi-source graph data while keeping sensitive information secure on local devices. Our results show that FedNGDBs can efficiently retrieve answers to cross-graph queries, making it a promising approach for LLMs and other applications that rely on efficient data retrieval mechanisms.

# 1 Introduction

Graph Databases (GDBs) excel at efficiently storing and managing highly interconnected data, leveraging their graph structure to efficiently manage complex relationships. This capability makes them indispensable for applications such as recommendation systems (Wang et al., 2019; Cao et al., 2019) and fraud detection (Prusti et al., 2021; Sadowski & Rathle, 2014), where the graph structure is critical. GDBs offer dynamic data models that adapt to evolving structures and deliver scalable performance for intricate queries. In the era of deep learning-based foundation models, such as large language models (LLMs), their importance has surged with the advent of Retrieval Augmented Generation (RAG), where agents utilize external GDBs, such as knowledge graphs (KGs) to enhance their retrieval capabilities (Gao et al., 2023; Lewis et al., 2020). This integration facilitates the creation of interactive natural language interfaces tailored to domain-specific applications, enabling more intuitive and accessible interaction with structured data and unlocking new possibilities for intelligent data-driven solutions (Matsumoto et al., 2024; Hussien et al., 2025; Jin et al., 2024).

However, traditional graph databases often suffer from two limitations: the ineffectiveness of free-text semantic search and graph incompleteness, which are prevalent issues in real-world knowledge graphs and other graph-structured data. Incompleteness leads to the exclusion of relevant results, as the graph database may not capture all the necessary relationships and connections between entities by traversing (Bordes et al., 2013; Galárraga et al., 2013). These limitations hinder their ability to fully support advanced retrieval tasks. To address this, neural graph databases (NGDBs) have recently been proposed by Besta et al. (2022); Ren et al. (2023). They integrate the adaptable structure of graph data models with the powerful processing capabilities of neural networks, allowing efficient storage, effective graph-structured data analysis, and flexible attribute representation (Zhang et al., 2024). NGDBs provide unified storage for diverse entries in an embedding space and a neural query engine searching for answers to complex queries from the unified storage (Ren et al., 2023). These databases unlock stronger capabilities for intelligent data exploration, allowing users to craft complex queries and make informed inferences with the help of advanced neural network techniques (Bai et al., 2025). Among these applications, complex query answering (CQA) is an important yet challenging task in graph reasoning and can be used to support various downstream tasks (Bai et al.; Ren et al., 2023). CQA aims to retrieve answers that satisfy given logical expressions (Hamilton et al., 2018; Ren et al., 2020), which are often defined in predicate logic forms with relation projection operations, existential quantifiers $\exists$, logical conjunctions $\wedge$, disjunctions $\vee$, etc. As shown in Figure 1, given a logical query $q$, our aim is to find all the research topic entities $V_?$ for which there exist Nobel Prize winners $V$ who were born in Germany and conducted studies in that specific field.

Although neural graph databases have achieved remarkable success in addressing complex query answering tasks, they are limited to utilizing a single central graph database and cannot be extended to multiple databases. As data plays an increasingly vital role, NGDBs have experienced rapid growth in scale and scope, aggregating knowledge from diverse domains. Consequently, constructing a graph database that includes all related entities and relations has become difficult and it is impractical to access a central database with all the data needed (Peng et al., 2021; Chen et al., 2021; Zhang et al., 2022). Collaborations between various NDGB holders are essential for answering more complicated queries. For example, as shown in Figure 1, there are multiple NGDBs with different domain knowledge. A complex query may consist of entities and relations from multiple NGDBs, preventing a local query answering model on a single database from answering those cross-graph queries. However, there are various reasons hindering the data sharing between NGDB holders. For example, the growing attention on privacy, regulations such as the General Data Protection Regulation

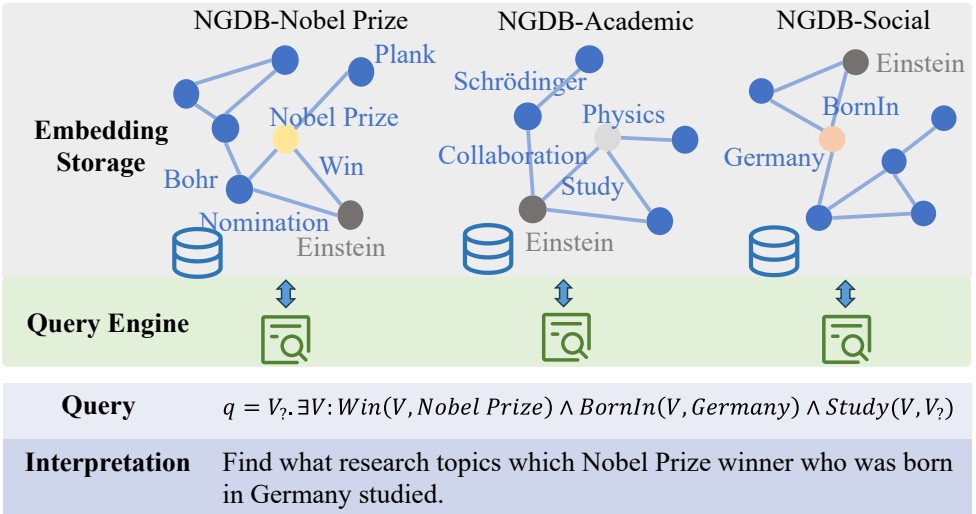

Figure 1: An example of cross graph queries on distributed neural graph databases. The relations and entities in a KG complex query can be from NGDBs which cannot be solved in a single local database.

(GDPR), and commercial interest between data holders, etc. Distributed database approaches exist (Nadal et al., 2021), but they fail to leverage NGDBs' neural capabilities fully.

Federated learning offers a potential solution, enabling collaborative model training across distributed participants without sharing raw data (McMahan et al., 2017; Yang et al., 2019). In federated knowledge graph embeddings, raw graph triplets are retained on local devices, participants train local models, and only gradients are transferred to learn a global graph embedding model (Chen et al., 2021; Zhang et al., 2022; Peng et al., 2021) under the protection of privacy preservation methods, such as homomorphic encryption (HE) (Paillier, 1999), secure multi-party computation (SMPC) (Mohassel & Zhang, 2017), and differential privacy (Dwork, 2008; Geyer et al., 2017). While federated learning has been widely applied in learning graph embeddings, recent studies have indicated that learned representations can still potentially leak privacy (Hu & Song, 2023; Duddu et al., 2020), where an attacker can infer sensitive information from the learned embeddings, besides, existing works only focus on learning high-quality representations for simple downstream tasks, such as knowledge graph completion (Chen et al., 2021; Zhang et al., 2022; Huang et al., 2022; Tang et al., 2023), and lacks the ability to reason over graphs and retrieving answers to complex queries.

To overcome these gaps, we propose to learn Federated Neural Graph DataBase (FedNGDB), a novel system that can reason over multi-source graphs avoiding sensitive raw data transmission and embedding exposure to safeguarding privacy. FedNGDB can be applied to different central complex query answering models. It leverages federated learning to train local query answering models in local NGDBs and align graph embeddings for global queries. Different from other federated graph embedding models, the FedNGDB server not only takes the responsibility of aggregating global models but also decomposing given complex queries to sub-queries to compute the query encoding and retrieve answers from distributed NGDBs under the protection of multi-party computation to avoid global model storage. Meanwhile, to better evaluate distributed NGDB systems' performance, we create a benchmark on three widely used datasets. We evaluate our proposed FedNGDBs on the benchmark, the experiment results show the effectiveness of retrieving answers to complex queries from multi-source graphs. We summarize our major contributions as follows:

- To the best of our knowledge, we are the first to extend federated graph embedding systems to complex query answering tasks, which is critical for graph holders' collaboration.

- Based on three public datasets, we propose a benchmark for evaluating the retrieval performance of distributed NGDB systems. The benchmark systematically evaluates the retrieval performance facing cross-graph queries.

- We propose FedNGDB, a federated neural graph database system that can retrieve answers of complex queries from distributed NGDBs with privacy preserved, solving the drawbacks of existing federated knowledge graph embeddings methods. Extensive experiments on three datasets demonstrate its high performance when facing cross-graph queries.

The rest of the paper is organized as follows. We review the related works in Section 2. Section 3 introduces the preliminary and definition of the distributed complex query answering systems. Section 4 introduces the framework of FedNGDB in detail. Section 5 evaluates the performance of FedNGDB on the benchmark based on three real-world datasets. Finally, we conclude our work in Section 6.

## 2 Related Work

### 2.1 Neural Graph Database

Neural Graph Databases (NGDBs) neutralize traditional GDBs' storage and query planning modules, aiming for stronger intelligent data exploration capabilities. Complex query answering (CQA) is a crucial task in NGDBs, which involves training a model to process and answer complex logical queries based on graph reasoning, a process known as query encoding. These methods represent complex queries into various structures and effectively search for answers among candidate knowledge graph entities. GQE (Hamilton et al., 2018), Q2B (Ren et al., 2020), and HypeE (Liu et al., 2021) encode queries to vectors, hyper-rectangle and hyperbolic embeddings, respectively. To support negation operators, various encoding methods are proposed. Q2P (Bai et al., 2022) and ConE (Zhang et al., 2021) use multiple vectors to represent complex queries. BetaE (Ren & Leskovec, 2020), GammaE (Yang et al., 2022), PERM (Choudhary et al., 2021) propose to use various probabilistic distributions to encode complex logic graph queries. Some methods, like LMPNN (Wang et al., 2022) CQD (Arakelyan et al., 2020), Var2Vec (Wang et al., 2023a) take pre-train embeddings on simple link prediction tasks and apply logic operators to answer complex queries. Meanwhile, neural structures are utilized to encode complex queries: BiQE (Kotnis et al., 2021) and KgTransformer (Liu et al., 2022) are proposed to use transformers, SQE (Bai et al.) applies sequential encoders, GNN-QE (Zhu et al., 2022) and StarQE (Alivanistos et al., 2021) use message-passing graph neural networks to encode queries, respectively. There are also encoding methods proposed to encode various types of knowledge graphs: NRN (Bai et al., 2023b) is proposed to encode numerical values, MEQE (Bai et al., 2023a) extends logical queries over events, states, and activities. P-NGDB (Hu et al., 2024) first extends the privacy preservation to neural graph databases.

While there are numerous existing complex query answering methods, these methods mainly focus on a single large graph. There are some methods proposed to reason over multi-view and temporal and varying graphs: MORA (Xi et al., 2022) ensembles multi-view knowledge graphs to scale up complex query answering, TTransE (Leblay & Chekol, 2018) and TRESCAL (Wang et al., 2015) can be applied on temporal knowledge graphs. However, these methods need raw data transmission and privacy protection is not considered. Conducting privacy-preserving complex query answering on multi-source knowledge graphs is still unexplored. With the growing attention on privacy and data protection, sensitive data cannot circulate freely among data holders, and complex query answering is forced to be conducted collaboratively on multiple knowledge graphs. Our research introduces federated learning to existing complex query answering so that we can apply reasoning over distributed NGDBs without raw data sharing.

### 2.2 Federated Knowledge Graph Embedding

In recent years, federated learning has emerged as a promising approach to address privacy and scalability concerns in machine learning. It allows data owners to participate in model co-construction without raw data transmission to reduce privacy leakage risks (McMahan et al., 2017; Yang et al., 2019) under the protection of privacy protection techniques such as differential privacy (DP) (Yang et al., 2019), homomorphic encryption

(HE) (Zhang et al., 2020), secure multi-party computation (SMPC) (Byrd & Polychroniadou, 2020). Various studies have been conducted to explore the potential of federated learning in different domains, such as recommender system (Yang et al., 2020), finance (Long et al., 2020) etc. Federated databases are proposed to manage distributed data allowing storing and querying databases with privacy preserved (Bater et al., 2017; Berger & Schrefl, 2008; Li, 2013). Although some federated graph databases are proposed to manage graph-based data (Nadal et al., 2021), the study for the latest neural graph databases are still be ignored.

Federated knowledge graph embedding is another related topic. It tries to represent entities and their semantic relations into embedding spaces. FedE (Chen et al., 2021) learns knowledge graph embeddings locally and aggregates all local models in a global server for higher representation quality. FedR (Zhang et al., 2022) proposes to learn representation with privacy-preserving relation aggregation to avoid privacy leakage risks in entity embedding and reduce communication costs. FedCKE (Huang et al., 2022), FedMKGC (Tang et al., 2023) extend federated learning to learn global representations from different domains and multilingual knowledge graphs. FedEC (Chen et al., 2022b) applies contrastive learning to tackle data heterogeneity in knowledge graphs. MaKEr (Chen et al., 2022a) and MorseE (Chen et al., 2022c) utilize meta-learning to transfer knowledge among knowledge graphs to train graph neural networks for unseen knowledge extrapolation and inductive learning. DP-FLames (Hu et al., 2023) quantifies the privacy threats and incorporates private selection in federated knowledge embeddings. FLEST (Wang et al., 2023b) decomposes the embedding matrix and enables the sharing of latent representations to reduce the risks of privacy leakage and communication costs, FedM (Hu et al., 2022a) splits the duty of aggregating entities and relations to reduce the risks of graph reconstruction attacks. FKGE (Peng et al., 2021) applies differential privacy and avoids the need for a central server. While existing federated knowledge graph embedding methods are proposed to distributedly learn high-quality representations with privacy preservation, there are some works indicate that the learned embeddings are informative and vulnerable to various privacy attacks (Hu et al., 2022b), even in federated scenarios (Hu & Song, 2024; Hu et al., 2023). Besides, these methods all lack the ability to answer complex queries on multi-source knowledge graphs which is critical for more complicated downstream tasks. Our research expands the simple federated knowledge graph embedding to answer complex queries on distributed knowledge graphs.

## 3 Preliminary and Problem Formulation

### 3.1 Preliminary

Following the general setting of federated knowledge graph embeddings, we denote a set of graph-structured data from various data owners as $\mathcal{G} = \{g_1, g_2, ..., g_N\}$, where $N$ is the total number of graphs. Data owners have their own graph data and cannot access to other's databases. Let $g_k = (\mathcal{V}_k, \mathcal{R}_k, \mathcal{T}_k)$ denotes the $k$-th graph in $\mathcal{G}$, where $\mathcal{V}_k$ denotes the set of vertices representing entities in the graph $g_k$, $\mathcal{R}_k$ denotes the set of relations, $\mathcal{T}_k$ denotes the set of triplets. Specifically, $\mathcal{T}_k = \{(v_h, r, v_t)\} \subseteq \mathcal{V}_k \times \mathcal{R}_k \times \mathcal{V}_k$ denotes there is a relation between $v_h$ and $v_t$, where $v_h, v_t \in \mathcal{V}_k$, $r \in \mathcal{R}_k$. We denote $\mathcal{V} = \cup_{k=1}^{N} \mathcal{V}_k$, $\mathcal{R} = \cup_{k=1}^{N} \mathcal{R}_k$, $\mathcal{T} = \cup_{k=1}^{N} \mathcal{T}_k$ as the set of vertices, relations, and triplets of all graph data, respectively.

### 3.2 Complex Logical Query

The complex logical query is defined in existential positive first-order logic form, consisting of various types of logic expressions like existential quantifiers $\exists$, logic conjunctions $\wedge$, and disjunctions $\vee$. In the logical expression, there is a set of anchor entities $V_a \in \mathcal{V}$ denotes given context, existential quantified variables $V_1, V_2, ..., V_k \in \mathcal{V}$, and a unique variable $V_?$ denotes our query target. The complex query intends to find the target answers $V_? \in \mathcal{V}$, such that there are $V_1, \cdots, V_k \in \mathcal{V}$ in the graph-structured data that can satisfy the given logical expressions simultaneously. Following the definition in Ren et al. (2020), the complex query expression can be converted to the disjunctive normal form (DNF) in the following:

$$q[V_?] = V_?.V_1, ..., V_k : c_1 \vee c_2 \vee ... \vee c_n$$
$$c_i = e_{i,1} \wedge e_{i,2} \wedge ... \wedge e_{i,m}, \tag{1}$$

where $e_{i,j}$ is the atomic logic expression, which can be the triplet $(V, r, V')$ denotes relation $r$ between entities $V$ and $V'$, $c_i$ is the conjunction of several atomic logic expressions $e_{i,j}$. $V, V'$ are either anchor entities or existentially quantified variables.

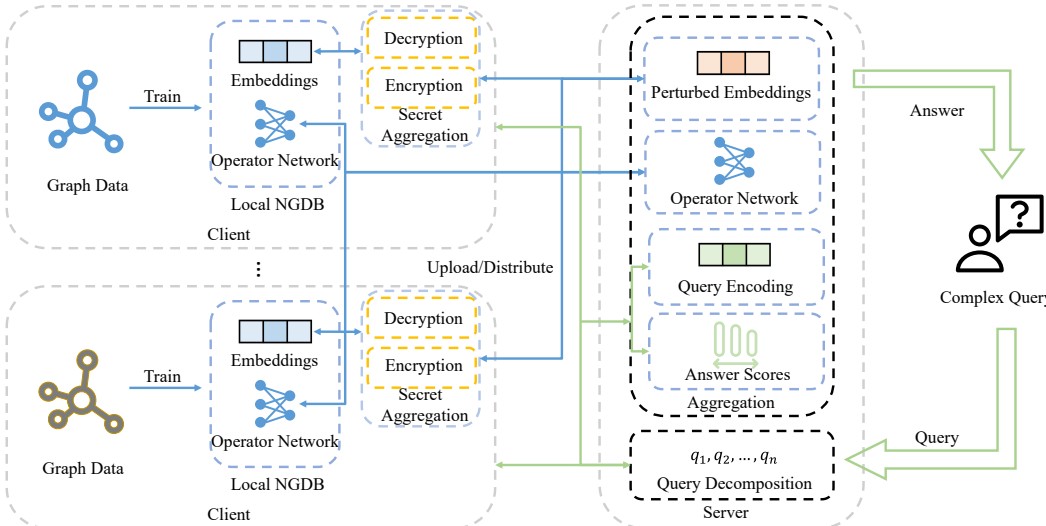

Figure 2: The training and retrieval process of FedNGDB. The blue line denotes the training process, and the green line denotes the retrieval process. In the training, clients' NGDB models are trained on respective graph-structured data. At each round, local NGDBs are aggregated at the server and updated using the global parameters. Among them, the embeddings are protected by secret aggregation so that the server can not access them. In the retrieval, each query is decomposed into sub-queries. Clients compute sub-query embeddings which the server is used to aggregate query embeddings. Answer scores are computed at clients and are aggregated at a server to retrieve answers.

### 3.3 Distributed Graph Set Query

Graph databases are owned by different data holders and cannot be shared directly with each other, therefore, a complex query $q$ can involve entities and relations from different graphs. We define those queries as follows:

**Definition 3.1 (Cross-graph Query)** *A complex query $q$ is a cross-graph query if there exists query answers $V_? \in \mathcal{V}$, such that there are $V_1, \cdots, V_k \in \mathcal{V}$ in the graph that can satisfy the given logical expressions and the atomic expressions in the query can not be found in a single graph.*

For example in figure 1, the entity "Physics" is the answer to the query $q$ because there exists an existentially quantified variable "Einstein" that can satisfy the logical expression. The query $q$ is a cross-graph query as the atomic expressions in the query are from different graph databases and can not be found in a single graph. For example, Win(Einstein, Nobel Prize) and BornIn(Einstein, Germany) are two atomic expressions from different graph databases. When the distributed graph databases face cross-graph queries, the answer can not be inferred from a single graph. Besides, We have the definition of in-graph query correspondingly:

**Definition 3.2 (In-graph Query)** *A complex query $q$ is an in-graph query if for all answers $V_? \in \mathcal{V}$ to the query, such that there are $V_1, \cdots, V_k \in \mathcal{V}$ in the graph that can satisfy the given logical expressions and the atomic expressions in the query are from a single graph database.*

These in-graph queries can retrieve answers according to one of the graphs in the graph set and can be solved with existing complex query answering models. However, in a distributed graph neural graph database system, the in-graph queries may retrieve more answers as more knowledge is provided.

### 3.4 Problem Formulation

Given a graph-structured data set $\mathcal{G} = \{g_1, ..., g_N\}$ with $N$ graphs. Every graph is owned by independent data holders and can not be shared to construct a unified graph database. We assume that the triples are sensitive as they describe the informative relations between entities while the index of entities and relations can be shared, which means that the triples in each graph database will stay private on local devices. The graphs in the $\mathcal{G}$ are related and have part of entities and relation overlapped. There are complex queries involving elements from the graph set $\mathcal{G}$ and can be classified as cross-graph queries and in-graph queries. We aim to construct a distributed neural graph database system to reason over multi-source graphs and retrieve answers to complex logical queries while keeping privacy preserved, especially the triple information indicating the relations between entities. To achieve this, we assume that there is an honest but curious server managing the federated neural graph database system. Because the learned embeddings are vulnerable to various privacy attacks, the embeddings cannot be exposed to the server and should be further protected before being transferred to the server.

## 4 Federated Neural Graph Databases

In this section, we introduce the learning and query retrieval process of our proposed FedNGDB.

### 4.1 Model Learning

We first introduce the training of FedNGDB. As shown in Figure 2, FedNGDB has a central server and a set of clients. Each client has a graph with overlapping entities of others. The server takes the responsibility of aggregating parameters and organizing the training and retrieval process. The clients train a local NGDB model based on their graph-structured data. According to the sensitivity of the parameters, we divide the query encoding methods into two parts: operator function with parameter $\Theta$ and entity embeddings $\mathbf{E}$. For the operator function, the client directly sends the parameter to the server for aggregation and receives the global function to update the local operator function, which is similar to FedAvg (McMahan et al., 2017). Therefore, in the following parts, we only introduce the entity embeddings aggregation in FedNGDB.

#### 4.1.1 Secret Aggregation

There are various techniques, like homomorphic encryption (HE) (Paillier, 1999), secure multi-party computation (MPC) (Mohassel & Zhang, 2017), and differential privacy (Dwork, 2008; Geyer et al., 2017) to protect the uploaded parameters, however the protection of aggregated global model are often ignored. Unfortunately, the global model is informative and vulnerable to privacy attacks (Zhang et al., 2022; Hu & Song, 2024). Hence we propose a secret aggregation applied in parameter aggregation which can prevent the global server from knowing the aggregated parameters using homomorphic encryption. Assume that at each client $i$, a parameter denoted as $\theta_i$ is uploaded to a server for aggregating global parameter $\theta$. The procedure of secret aggregation is described in Algorithm 1. In the beginning, each client $C_i$ randomly generates perturbed parameters $\theta_i^r$ and shares them with other clients with encryption (Shown in Appendix A). After the sharing, each client has a set of parameters $\{\theta_1^r, \theta_2^r, \cdots, \theta_n^r\}$. In the training process, for each client, $C_i$ uploads perturbed parameter $(\theta_i + \theta_i^r)$ to the server under the protection of homomorphic encryption. The server collects all perturbed parameters from clients, computes aggregated perturbed parameter $\theta^r$, and sends it back to all clients. Finally, after receiving the perturbed parameters, the clients first decrypt the parameters and can compute aggregated parameter $\theta$ by removing the perturbed parameters and preventing exposing it to the server.

Besides, the parameters are also protected by the differential privacy to prevent the potential privacy leakage (Zhu et al., 2019), we use local differential privacy (LDP) to the local gradients to further improve the difficulty to recover sensitive information from transmission. As proposed in Yi et al. (2021), we clip the local gradients based on the $L_\infty$ norm with the threshold $\mathcal{C}$ and apply zero-mean Laplacian noise with strength $\lambda$, the upper bound of the privacy budget $\epsilon$ is $\frac{2\mathcal{C}}{\lambda}$. The detailed discussion can be found in Appendix B.3.

### 4.1.2 Model Training

Similar to Chen et al. (2021), the server constructs a set of mapping matrices $\{\mathbf{M}^i \in \{0,1\}^{n \times n_i}\}_{i=1}^N$ and existence vectors $\{\mathbf{v}^i \in \{0,1\}^{n \times 1}\}_{i=1}^N$ to denote the entities in each client, where $n$ is the number of all unique entities in KG set and $n_i$ is the number of entities from client $C_i$. $\mathbf{M}^i_{m,n} = 1$ if the $m$-th entity in entity table corresponds to the $n$-th entity from client $C_i$. $\mathbf{v}^i_m = 1$ indicates that the $m$-th entity in entity table exists in client $C_i$.

FedNGDB performs secret aggregation for entity embeddings. First, the client $C_i$ will randomly initialize the local entity embeddings $\mathbf{E}^i_0 \in \mathbb{R}^{n_i \times d}$ and perturbation embeddings $\mathbf{E}^i_r \in \mathbb{R}^{n_i \times d}$. Every client will share the permutation embeddings with all clients. At round $t$, the server will select part of the clients $\mathbf{C}$ participating in the training. After local training of CQA on respective local graphs, client $C_i$ sends perturbed local entity embeddings $\mathbf{E}^i_t + \mathbf{E}^i_r$ to the server. The server will aggregate the entity embeddings using perturbed local embeddings:

$$\mathbf{E}^r_{t+1} \leftarrow \left(\mathbb{1} \oslash \sum_{i \in \mathbf{C}} \mathbf{v}^i\right) \otimes \sum_{i \in \mathbf{C}} \mathbf{M}^i (\mathbf{E}^i_{t+1} + \mathbf{E}^i_r), \tag{2}$$

where $\mathbb{1}$ denotes all-one vector, $\oslash$ denotes element-wise division for vectors and $\otimes$ denotes element-wise multiply with broadcasting. After aggregation, the server sends the aggregated entity embeddings back to all clients, and the client $C_i$ receives:

$$\mathbf{E}^{r,i}_{t+1} \leftarrow \mathbf{M}^{i\top} \mathbf{E}^r_{t+1}, \tag{3}$$

and the client $C_i$ can compute and update the local entity embeddings as:

$$\mathbf{E}^i_{t+1} \leftarrow \mathbf{E}^{r,i}_{t+1} - \mathbf{M}^{i\top} \left(\mathbb{1} \oslash \sum_{j \in \mathbf{C}} \mathbf{v}^j\right) \otimes \sum_{j \in \mathbf{C}} \mathbf{M}^j \mathbf{E}^j_r. \tag{4}$$

After secret aggregation, the entity embeddings of all clients are shared without exposing the sensitive information to the server. Besides the training of entity embeddings, the operator networks are trained and aggregated using FedAvg (McMahan et al., 2017), and the detailed descriptions of the FedNGDB are shown in Algorithm 2.

## 4.2 Query Retrieval

After training, the server in FedNGDB manages the process of retrieving answers to complex queries. The server first tries to arrange related clients to encode the coming queries and retrieves answers from all local graph databases based on the encoding.

### 4.2.1 Query Encoding

Query encoding methods commonly represent queries to embeddings and retrieve answers according to scoring functions where similarity functions are widely used. FedNGDB encodes queries and treats two types of queries differently. For in-graph queries, as these queries only involve a single graph data, FedNGDB can directly encode queries using corresponding local complex query answering models. For cross-graph queries, as the query involves entities from multiple graphs, the server will take the responsibility to plan the entire encoding process: First, the server will decompose the query to atomic expressions and send each expression to corresponding local graphs. The clients encode received atomic queries using their own local CQA models and send back the results to the server. The server collects all the encoding results and uses global operator function models to compute the representations of the queries. The query is iteratively updated by communicating between the server and clients until the original queries are encoded.

### 4.2.2   Answer Retrieval

Because the entity embeddings are not stored in the central server, we can only retrieve answers from all distributed local graphs after encoding the queries. Given a query encoding $q$, we score all the candidate entities at each local graph database, at client $C_i$:

$$\mathbf{S}_q^i \leftarrow f_s^i(\mathcal{V}_i, q) \in \mathbb{R}^{n_i \times 1}, \tag{5}$$

where $f_s^i$ is a score function in the client $C_i$. Then the score will be uploaded to the server to aggregate a score table for all unique entities in the graph sets:

$$\mathbf{S} \leftarrow \left( \mathbb{1} \oslash \sum_{i=1}^{N} \mathbf{v}^i \right) \otimes \sum_{i=1}^{N} \mathbf{M}^i \mathbf{S}^i. \tag{6}$$

The final answers to the queries are retrieved globally from the graph database set according to the score table.

## 5   Experiments

In this section, we create a benchmark of distributed graph complex logical query answering problems for distributed neural graph databases and evaluate our proposed FedNGDB's performance on the benchmark.

### 5.1   Datasets and Experiment Setting

We introduce the detailed information of our used datasets and the setting of our experiments.

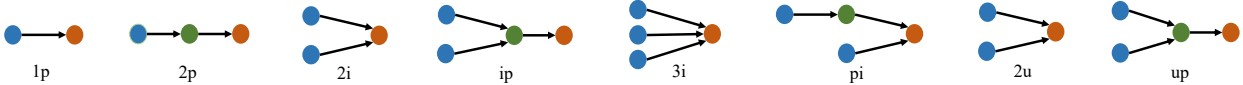

Figure 3: The query structures used for evaluation in the experiments. Naming for each query structure is provided under each subfigure, for brevity, the p, i, and u represent the projection, intersection, and union operations respectively.

### 5.1.1   Datasets

In our experiment, following previous work, we use the three commonly used knowledge graphs in central graph reasoning tasks as graph-structured data: FB15k (Bollacker et al., 2008; Bordes et al., 2013), FB15k-237 (Toutanova & Chen, 2015), and NELL995 (Carlson et al., 2010) to construct the distributed query answering benchmark, which can make better comparison to existing works. In each dataset, there are vertices describing entities and edges describing relations. To evaluate the distributed complex query, we follow the common settings of federated knowledge graph emebddings (Chen et al., 2021; Zhang et al., 2022), conducting experiments assuming having 3 and 5 clients in each federated neural graph database system, respectively. For more clients, we also conduct simple exeperiments as shown in Appendix C.1. We randomly select relations into clients and split triples into clients according to selected relations as a local graph database. The triplets in each local graph are separated into training, validation, and testing with a ratio of 8:1:1 respectively. Following previous works (Ren et al., 2020), we construct training graph $\mathcal{G}_{train}$, validation graph $\mathcal{G}_{val}$, and test graph $\mathcal{G}_{test}$ in each client by training edges, training+validation edges, and training+validation+testing edges, respectively. The detailed statistics are listed in the table 1. #Clients denotes the number of clients, #Nodes, #Relations, #Edges denote the average number of nodes, relations, and edges in each client respectively.

Table 1: The statistics of three datasets used for experiments.

| Graphs | #Clients | #Nodes | #Relations | #Edges |
|---|---|---|---|---|
| FB15k-237 | 3 | 13,651 | 79 | 103,359 |
| | 5 | 12,639 | 47.4 | 62,015 |
| FB15k | 3 | 14,690 | 448.3 | 197,404 |
| | 5 | 14,279 | 269 | 118,442 |
| NELL995 | 3 | 40,204 | 66.7 | 47,601 |
| | 5 | 28,879 | 40 | 28,560 |

Table 2: The statistics of queries sampled from three datasets used for experiments.

| Graphs | #C | In-graph | | | Cross-graph |
| | | Train. | Valid. | Test. | Test. |
|---|---|---|---|---|---|
| FB15k-237 | 3 | 317,226 | 11,528 | 11,539 | 32,573 |
| | 5 | 180,552 | 6,619 | 6,673 | 31,469 |
| FB15k | 3 | 592,573 | 19,206 | 19,267 | 53,660 |
| | 5 | 344,418 | 11,409 | 11,437 | 53,154 |
| NELL995 | 3 | 208,070 | 8,810 | 8,750 | 24,954 |
| | 5 | 117,231 | 5,177 | 5,118 | 24,237 |

### 5.1.2 Query Sampling

Following previous work (Hamilton et al., 2018; Bai et al., 2023b), we evaluate the complex logical query answering performance on the following eight general query types with abbreviations of $1p, 2p, 2i, ip, 3i, pi, 2u$, and $up$. As shown in figure 3, each subgraph denotes one query type, where each edge represents either a projection or a logical operator, and each node represents either a set of entities, the anchor entities and relations are to be specified to instantiate logical queries. We use the sampling method commonly used in previous works Bai et al.; Ren et al. (2020) to randomly sample complex queries from graphs. We randomly sample two sets of queries from the graph sets: in-graph queries and cross-graph queries to evaluate local and global answer retrieval performance. For local model evaluation, we first obtain training, validation, and testing queries from the formerly constructed local graph databases respectively. Then for the training queries, we conduct a graph search to find corresponding training answers on the local training graph. For the validation queries, we search for the answers on both the training graph and the validation graph and only use those queries that have different numbers of answers on the two graphs. For the testing queries, we use those queries that have different answers on the testing graph from answers on the training graph and validation graph. For global model evaluation, we construct global training, validation, and testing graphs using all local graphs, and sample testing queries with atomic expressions from different local graphs, finally we search for answers on three global graphs and only use those queries that have different answers on the testing graph from other two graphs. We collect statistics of complex queries in three datasets and the statistics are shown in Table 2. The number of in-graph queries is the average number of the client's local queries.

### 5.1.3 Baselines

We can use various existing query encoding methods as our local base model, to evaluate the effectiveness and generalization ability of our proposed FedNGDB, we select three commonly used complex encoding methods GQE (Hamilton et al., 2018), Q2P (Bai et al., 2022), Tree-LSTM (Bai et al.) as our base model. GQE is a graph query encoding model that encodes a complex query into a vector in embedding space; Q2P represents complex queries using multiple vectors; Tree-LSTM recursively represents complex queries and treats all operations, entities, and relations as tokens.

Table 3: The retrieval performance of distributed neural graph databases when there are 3 clients. The average results of HR@3 and MRR of all clients are reported. The best results are underlined. The best results of distributed models are in bold.

| Graph | Setting | GQE | | | | Q2P | | | | Tree-LSTM | | | |
|---|---|---|---|---|---|---|---|---|---|---|---|---|---|
| | | In-graph | | Cross-graph | | In-graph | | Cross-graph | | In-graph | | Cross-graph | |
| | | HR@3 | MRR | HR@3 | MRR | HR@3 | MRR | HR@3 | MRR | HR@3 | MRR | HR@3 | MRR |
| FB15k-237 | Local | 12.64 | 12.03 | - | - | 14.55 | 13.63 | - | - | 13.32 | 12.73 | - | - |
| | Central | 13.13 | 12.39 | 13.03 | 12.28 | 14.93 | 14.66 | 15.02 | 14.81 | 13.28 | 12.61 | 13.36 | 12.91 |
| | FedE | **13.72** | **13.23** | **12.74** | **11.63** | 14.82 | 14.27 | 14.79 | 13.93 | 13.12 | 12.23 | **12.62** | **12.08** |
| | FedR | 12.89 | 11.98 | - | - | 14.32 | 14.23 | - | - | **13.92** | **12.92** | - | - |
| | FedNGDB | 13.54 | 12.43 | 12.63 | 11.32 | **15.32** | **14.32** | 14.83 | 14.11 | 12.93 | 12.11 | 12.55 | 11.96 |
| FB15k | Local | 22.05 | 18.21 | - | - | 24.32 | 22.64 | - | - | 22.87 | 20.51 | - | - |
| | Central | 29.53 | 25.65 | 30.21 | 25.33 | 38.62 | 34.14 | 38.03 | 34.36 | 38.87 | 35.86 | 37.97 | 36.13 |
| | FedE | 24.31 | 26.74 | **27.95** | **25.21** | 43.68 | 39.62 | 39.72 | 35.95 | 34.27 | 30.18 | **31.19** | 26.03 |
| | FedR | 20.29 | 18.61 | - | - | 25.32 | 22.71 | - | - | 23.64 | 20.97 | - | - |
| | FedNGDB | **25.63** | **26.87** | 24.77 | 25.17 | **44.02** | **39.27** | **40.27** | **36.31** | **34.85** | **33.83** | 31.80 | **28.99** |
| NELL995 | Local | 11.85 | 11.03 | - | - | 15.86 | 13.02 | - | - | 13.85 | 13.85 | 12.94 | - |
| | Central | 12.87 | 11.95 | 13.06 | 12.46 | 16.74 | 14.82 | 16.42 | 15.63 | 15.41 | 14.23 | 16.27 | 15.83 |
| | FedE | 13.29 | 12.72 | 12.46 | 11.82 | **17.23** | 14.12 | **16.28** | 14.01 | 14.27 | 13.81 | 14.18 | 13.71 |
| | FedR | 12.01 | 11.23 | - | - | 16.04 | 13.26 | - | - | 12.48 | 11.67 | - | - |
| | FedNGDB | **14.21** | **13.27** | **13.76** | **12.67** | 16.62 | **15.28** | 16.27 | **16.23** | **16.28** | **15.38** | **16.09** | **15.27** |

Table 4: The retrieval performance of distributed neural graph databases when there are 5 clients. The average results of HR@3 and MRR of all clients are reported. The best results are underlined. The best results of distributed models are in bold.

| Graph | Setting | GQE | | | | Q2P | | | | Tree-LSTM | | | |
|---|---|---|---|---|---|---|---|---|---|---|---|---|---|
| | | In-graph | | Cross-graph | | In-graph | | Cross-graph | | In-graph | | Cross-graph | |
| | | HR@3 | MRR | HR@3 | MRR | HR@3 | MRR | HR@3 | MRR | HR@3 | MRR | HR@3 | MRR |
| FB15k-237 | Local | 11.44 | 10.65 | - | - | 14.65 | 13.8 | - | - | 11.23 | 10.37 | - | - |
| | Central | 13.72 | 12.87 | 12.99 | 12.74 | 15.93 | 14.62 | 15.74 | 15.23 | 13.62 | 12.54 | 11.86 | 11.53 |
| | FedNGDB | **12.42** | **11.60** | **11.20** | **10.79** | **16.13** | **15.78** | **15.28** | **14.91** | **12.48** | **11.91** | **11.49** | **11.02** |
| FB15k | Local | 19.83 | 17.51 | - | - | 36.10 | 35.04 | - | - | 20.03 | 18.62 | - | - |
| | Central | 20.58 | 20.14 | 21.21 | 21.02 | 42.56 | 41.87 | 39.57 | 39.46 | 24.88 | 24.76 | 24.93 | 24.72 |
| | FedNGDB | **21.40** | **20.83** | **20.71** | **19.94** | **40.81** | **37.96** | **38.56** | **35.73** | **24.59** | **23.75** | **23.85** | **22.90** |
| NELL995 | Local | 10.48 | 10.09 | - | - | 15.26 | 14.37 | - | - | 14.52 | 13.89 | - | - |
| | Central | 14.56 | 14.28 | 13.75 | 13.42 | 15.63 | 15.14 | 15.27 | 14.75 | 16.47 | 15.92 | 15.74 | 14.47 |
| | FedNGDB | **13.79** | **13.27** | **12.74** | **12.18** | 15.44 | **15.81** | **15.28** | **14.24** | **15.68** | **14.28** | **14.57** | **12.89** |

To the best of our knowledge, there are no existing federated complex query answering methods but several federated knowledge graph embedding methods, therefore, we choose to compare our methods with FedE (Chen et al., 2021) and FedR (Zhang et al., 2022), two commonly used federated knowledge graph embedding methods as baselines. FedE aggregates both entity embeddings and relation embeddings in a server, while FedR only aggregates relation embeddings for privacy concerns and communication efficiency. We utilize these two methods with slight modifications to train a global complex query answering model: FedE aggregates all query encoding parameters and FedR aggregates relation embeddings and query encoding networks. Besides, we also compare our FedNGDB with local and central settings. These two baselines represent the lower bound and upper bound, respectively, in terms of the amount of information available to the training system. In the local setting, there is no collaboration between clients, while in the central setting, all distributed graphs in the graph set are aggregated for a global graph for training, we sample complex queries from global training, and validation graphs for training and validation.

If there is no further statement, we use the following implementation settings in the experiments. We tune hyper-parameters on the validation local queries for the base query encoding methods and set the dimension of entities and relations as 400 for all models for fair comparison and use AdamW (Loshchilov & Hutter, 2018) as optimizer. We set the gradient clip threshold $C = 0.1$ and Laplacian noise scale $\lambda = 0.2$ to achieve 1-DP.

### 5.1.4 Evaluation Metrics

Following the previous work (Bai et al., 2023b), we evaluate the generalization capability of models by calculating the rankings of answers that cannot be directly retrieved from an observed graph. Given a testing query $q$, the training, validation, and public testing answers are denoted as $M_{train}$, $M_{val}$, and $M_{test}$, respectively. We evaluate the quality of retrieved answers using Hit ratio (HR) and Mean reciprocal rank (MRR). HR@K metric evaluates the accuracy of retrieval by measuring the percentage of correct hits among the top K retrieved items. The MRR metric evaluates the performance of a ranking model by computing the average reciprocal rank of the first relevant item in a ranked list of results. The metric can be defined as:

$$\text{Metric}(q) = \frac{1}{|M_{test}/M_{val}|} \sum_{v \in M_{test}/M_{val}} m(rank(v)), \tag{7}$$

$m(r) = \mathbf{1}[r \leq K]$ if the metric is HR@K and $m(r) = \frac{1}{r}$ if the metric is MRR. Higher values denote better reasoning performance. We train local models at each client by using the in-graph training queries and tune hyper-parameters using the validation queries. The evaluation is then finally conducted on the testing queries, including the evaluation of in-graph queries on local query encoding models and cross-graph queries on the global federated neural graph database system, respectively.

## 5.2 Performance Evaluation

We evaluate FedNGDB's complex query answering performance on three datasets and compare it to other baselines. We apply FedNGDB on three base query encoding models and evaluate the average retrieval performance on various queries. The results are summarized in Table 3 and Table 4. Table 3 reports the retrieval performance of various distributed graph complex query answering models when there are 3 clients. For each model, we evaluate performance facing in-graph queries and cross-graph queries, respectively. For in-graph queries, the average scores of all clients are reported. We report results in HR@3 and MRR which higher scores indicate better performance. The best results are underlined. The best results of distributed models are in bold. As shown in Table 3, our proposed methods can effectively retrieve complex query answers from distributed graph databases. In comparison to local settings, we can see that FedNGDB can utilize all participated local graph databases and performs better facing in-graph queries. For example, GQE model with FedNGDB can achieve 14.21 HR@3 on average while can only reach 11.85 without collaboration. Besides, in comparison to other federated knowledge graph embedding methods, our proposed FedNGDB can reach comparable performance in both in-graph queries and cross-graph queries without exposing sensitive entity embeddings to the server. For example, FedNGDB achieves the best performance in cross-graph queries in more than half of datasets and base query encoding models.

In Table 4, we present the performance of FedNGDB when there are 5 clients. We compare the performance with local training without collaboration to demonstrate the influence of client numbers. As shown in the table, FedNGDB performs well compared to complex query answering models without collaboration, there are performance improvements in all datasets after applying FedNGDB to various base query encoding models, demonstrating that FedNGDB can utilize the intrinsic information in the distributed knowledge sets. The collaboration allows FedNGDB to reason over various logical paths to improve performance.

## 5.3 Query Types

FedNGDB can globally reason over distributed graphs and retrieve answers to cross-graph queries. To evaluate FedNGDB's performance on various types of complex queries, we conduct experiments to evaluate the retrieval performance of FedNGDB and compare it to central learning on FB15k-237 when there are 3 clients. Figure 4(a) shows the FedNGDB-GQE's performance facing cross-graph queries. As we can see from the figure, FedNGDB performs well on most various types of queries compared to the central model. For example, on query types '2i' and 'pi', FedNGDB can reach more than 90% MRR compared to the central model.

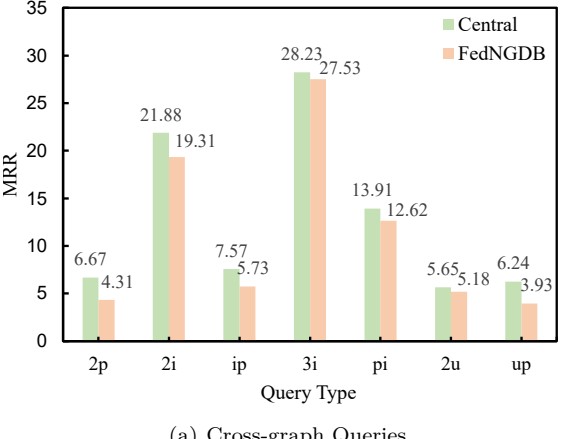
(a) Cross-graph Queries

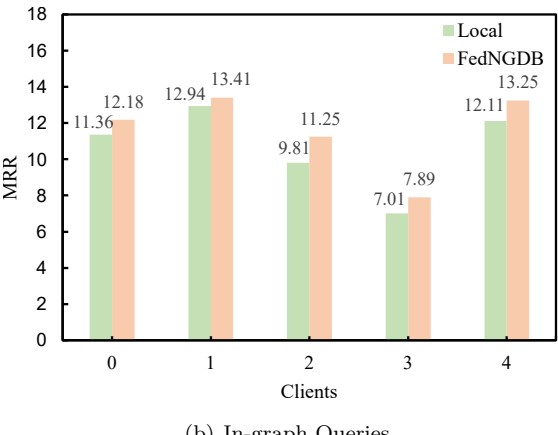
(b) In-graph Queries

Figure 4: The evaluation results of FedNGDB-GQE facing cross-graph queries on FB15k-237 (subfigures (a)). The evaluation results of FedNGDB-GQE facing in-graph queries on FB15k-237 (subfigures (c)).

## 5.4 Local Influence

Because the in-graph queries can be processed by a single local neural graph database, in this part, we evaluate the performance of FedNGDB on these queries to assess the influence of FedNGDB on local queries. We conduct experiments on FB15k-237 and the number of clients is 5. We evaluate the performance of FedNGDB based on GQE and compare the model with no collaboration. The results are summarized in the Figure 4(b). As shown in the figure, although each client has different performance due to the triplets correlation being different in each sub-dataset, FedNGDB can improve all clients' performance compared to the model without collaboration.

## 6 Conclusion

In this work, we present a federated neural graph database, FedNGDB to reason over distributed knowledge sets with privacy preservation, allowing graph database holders to collaboratively build a distributed graph reasoning system without sharing raw data. We define the distributed graph complex logical query answering problem. To solve the problem, we propose secret aggregation for federated learning where the aggregated parameters can be kept secret to the server. Besides, we design a distributed query retrieval process for answering queries from distributed graph database sets to protect clients' privacy. To evaluate FedNGDB model performance, we construct a benchmark based on three commonly used knowledge graph complex query answering datasets: FB15k-237, FB15k, and NELL995. Extensive experiments on the benchmark demonstrate the effectiveness of our proposed FedNGDB. FedNGDB can retrieve answers given a query while keeping the sensitive information secret at local graph databases. In the future, we aim to propose new methods for better answering complex queries by exploiting intrinsic information in the distributed neural graph databases.

## Acknowledgement

The authors of this paper were supported by the ITSP Platform Research Project (ITS/189/23FP) from ITC of Hong Kong, SAR, China, and the AoE (AoE/E-601/24-N), the RIF (R6021-20) and the GRF (16205322) from RGC of Hong Kong,SAR, China.

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

# A Parameter Sharing

In this section, we provide an example of sharing secrets between clients. The parameters can be shared under the protection using various encryption methods, for example, the commonly used is Diffie–Hellman key exchange (Diffie & Hellman, 2022) shown as follows, we consider the sharing process between two clients:

- Client A and Client B publicly agree to use a modulus $p$ and base $g$, $p$ is a prime.

- Client A chooses a secret integer $a$, then sends Client B $m_A = g^a \mod p$.

- Client B chooses a secret integer $b$, then sends Client A $m_B = g^b \mod p$.

- Client A computes $s = m_B^a \mod p$.

- Client B computes $s = m_A^b \mod p$.

After D-H key exchange, Client A and B share a secret $s = g^{ab} \mod p$. The secret $s$ can be used as encryption to share sensitive information between clients.

# B Alogrithm

## B.1 Secret Aggregation

We present the pseudo-code of secret aggregation in Algorithm 1.

---
**Algorithm 1:** Secret Aggregation

---
**Require:** $n$ clients $C_1, C_2, \ldots, C_n$, client $C_i$ has parameter $\theta_i$
  Each client $C_i$ generates random parameter $\theta_i^r$
  Transmit $\theta_i^r$ to all other clients with encryption, each
  client has a set of parameters $\{\theta_1^r, \theta_2^r, \cdots, \theta_n^r\}$

  **Client $C_i$:**
  Upload perturbed parameter $(\theta_i + \theta_i^r)$ to server with HE Encryption.
  Receive encrypted parameters $\sum_{j=1}^{n}(\theta_j + \theta_j^r)$ from server and HE decryption.
  Compute encrypted parameters $\theta = [\sum_{j=1}^{n}(\theta_j + \theta_j^r) - \sum_i^n \theta_i^r]/n$

  **Server:**
  **for** $i = 1, \cdots, n$ **do**
    Receive encrypted parameters $(\theta_i + \theta_i^r)$ from client $C_i$
  **end for**
  Compute encrypted parameters $\theta^r = \sum_{j=1}^{n}(\theta_j + \theta_j^r)$
  Send encrypted parameters $\theta^r$ to all clients

---

## B.2 FedNGDB Framework

We present the pseudo-code of FedNGDB in Algorithm 2.

## B.3 Differential Privacy Setting

Differential privacy represents a mathematical framework for quantifying and limiting the disclosure of private information in statistical databases (Dwork, 2008). In federated learning, differential privacy enables collaborative model training across decentralized data sources while providing formal privacy guarantees. By injecting calibrated noise into the training process, these techniques ensure that individual contributions

---

**Algorithm 2:** FedNGDB Framework

---

**Require:** The number of clients $N$; The faction of clients
   selected in each round $F$;

**Client $C_i$:**
Client $C_i$ initialize entity embeddings $\mathbf{E}^i$ and perturbation embeddings $\mathbf{E}^i_r$.
Share $\mathbf{E}^i_r$ with other clients with encryption
Receive and decryption to get $\{\mathbf{E}^1_r, \cdots, \mathbf{E}^N_r\}$
Upload $\mathbf{E}^i$ to server for secret aggregation, receive $\mathbf{E}^i_0$

**Server:**
Server constructs permutation matrices $\{M_i\}_{i=1}^N$, and existence vectors $\{v_i\}_{i=1}^N$ and initialize operator
networks $\Theta_0$, distribute to all clients.
**for** $t = 0, 1, 2, \cdots$ **do**
   Server distributes operator networks to each client.
   $\mathcal{C}_t \leftarrow$ Randomly select client set with $N \times F$ clients.
   **for** $C_i \in \mathcal{C}_t$ **in parallel do**
      $(\mathbf{E}^i_r + \mathbf{E}^i_{t+1}), \Theta^i_{t+1} \leftarrow \mathbf{ClientUpdate}(C_i, \mathbf{M}^{i\top}\mathbf{E}^r_t, \Theta_t)$
   **end for**
   $\mathbf{E}^r_{t+1} \leftarrow \left(\mathbb{1} \oslash \sum_{i \in \mathbf{C}_t} \mathbf{v}^i\right) \otimes \sum_{i \in \mathbf{C}_t} \mathbf{M}^i (\mathbf{E}^i_{t+1} + \mathbf{E}^i_r)$
   $\Theta_{t+1} \leftarrow 1/|\mathcal{C}_t| \sum_{i \in \mathcal{C}_t} \Theta^i_t$
**end for**

**ClientUpdate$(C_i, \mathbf{E}^r, \Theta)$:**
$\mathbf{E} \leftarrow \mathbf{E}^r - \mathbf{M}^{i\top}\left(\mathbb{1} \oslash \sum_{j \in \mathbf{C}} \mathbf{v}^j\right) \otimes \sum_{j \in \mathbf{C}} \mathbf{M}^j \mathbf{E}^j_r$
**for** $e = 1, \cdots, E$ **do**
   $\mathbf{E}, \Theta \leftarrow \mathbf{LocalUpdate}(\mathbf{E}, \Theta)$
**end for**
**return** $\mathbf{E} + \mathbf{E}^i_r, \Theta$

---

cannot be reliably identified or extracted from the global model (Geyer et al., 2017). The privacy-utility tradeoff can be tuned via the privacy budget parameter $\epsilon$, allowing practitioners to balance learning performance against disclosure risk. The formal definition of $\epsilon$-differential privacy ($\epsilon$-DP) is as follows (Dwork, 2006): A randomized algorithm $M$ satisfies $\epsilon$-differential privacy if for all neighboring datasets $D_1$ and $D_2$ that differ in at most one record, and for all possible outputs $S \subseteq Range(M)$:

$$Pr[M(D_1) \in S] \leq e^\epsilon Pr[M(D_2) \in S]$$

Where $\epsilon$ is the privacy parameter that quantifies the privacy loss.

In the former part, we introduce Laplacian noise to the local gradients to meet the guarantee of $\epsilon$-DP for in-graph query in local neural graph databases. However, for cross-graph query, we compute the score on each graph database for each query. The Post-Processing Theorem (Dwork et al., 2014) which states the composition of a data-independent mapping with an $\epsilon$-DP algorithm $M$ is also $\epsilon$-DP. Since the ranking operation of FedNGDB is a post-processing step applied to the outputs $(M_1(D), \cdots, M_n(D))$ from each local databases and the overlapped candidate answers share same scores, by the post-processing theorem of differential privacy, the final ranked list maintains the same privacy guarantee of $\epsilon$-DP.

## C   Auxiliary Experiments

Here we present some auxiliary experiments to further evaluate the performance of FedNGDB.

## C.1 More Clients

In the former experiments, we evaluate FedNGDB's performance when there are 3 or 5 clients in the federated system. To further evaluate models' performance when there are more clients. We split the graph-structured data into 10 subgraphs and evaluate the retrieval performance of FedNGDB using GQE as the base model. As shown in Table 5, FedNGDB can still improve the retrieval performance compared to local training when there are more clients participating in the distributed system, demonstrating the effectiveness of FedNGDB.

Table 5: The performance of GQE when #C=10.

| Graph | Setting | FB15k-237 | | | | FB15k | | | | NELL995 | | | |
|---|---|---|---|---|---|---|---|---|---|---|---|---|---|
| | | In-graph | | Cross-graph | | In-graph | | Cross-graph | | In-graph | | Cross-graph | |
| | | HR@3 | MRR | HR@3 | MRR | HR@3 | MRR | HR@3 | MRR | HR@3 | MRR | HR@3 | MRR |
| GQE | Local | 8.96 | 8.47 | - | - | 13.46 | 13.04 | - | - | 8.36 | 7.83 | - | - |
| | Central | 13.15 | 12.76 | 13.24 | 12.89 | 19.76 | 19.43 | 20.27 | 19.87 | 13.76 | 13.28 | 13.53 | 13.27 |
| | FedNGDB | 10.67 | 10.17 | 10.23 | 10.04 | 15.39 | 14.98 | 14.86 | 14.45 | 12.93 | 12.11 | 12.55 | 11.96 |

## C.2 Relation Overlap

In the experiments, we evaluate the retrieval performance when there are no overlap relations between local graph databases, however, various graph databases can have shared relations in real life, to evaluate the performance in such a scenario, we evaluate the FedNGDB with GQE's retrieval performance. The graph-structured data is randomly split into 3 subgraphs. The results are shown in Table 6, showing that FedNGDB can successfully retrieve answers from distributed graph databases.

Table 6: The MRR of GQE when relation overlapped.

| Graph | Setting | FB15k-237 | | | | FB15k | | | | NELL995 | | | |
|---|---|---|---|---|---|---|---|---|---|---|---|---|---|
| | | In-graph | | Cross-graph | | In-graph | | Cross-graph | | In-graph | | Cross-graph | |
| | | HR@3 | MRR | HR@3 | MRR | HR@3 | MRR | HR@3 | MRR | HR@3 | MRR | HR@3 | MRR |
| GQE | Local | 10.48 | 10.22 | - | - | 20.76 | 20.21 | - | - | 10.03 | 9.64 | - | - |
| | Central | 13.47 | 13.27 | 13.86 | 13.54 | 30.74 | 30.46 | 30.27 | 30.16 | 13.82 | 13.68 | 13.56 | 13.12 |
| | FedNGDB | 11.96 | 11.42 | 11.67 | 11.43 | 23.89 | 22.47 | 23.22 | 22.89 | 11.68 | 11.36 | 12.03 | 11.89 |

## C.3 Convergence Rate

We evaluate the convergence speed of three federated frameworks. The results are presented by the average number of communication round ratios relative to FedE. As shown in Table 7, FedNGDB's convergence speed is faster than FedR while slightly slower than FedE, demonstrating that our FedNGDB can protect stronger protection while remaining competitive efficiency.

Table 7: The statistics of communication rounds .

| Setting | FedE | FedR | FedNGDB |
|---|---|---|---|
| Relative Rounds to FedE | 1.00 | 1.32 | 1.09 |

