# OpenReview forum: "Learning Federated Neural Graph Databases for Answering Complex Queries from Distributed Knowledge Graphs"
_TMLR — Accepted by TMLR_

### Review · Reviewer_RKTW · 2025-04-06

**Summary Of Contributions:**

The manuscript addresses the limitations of current NGDBs, particularly their inability to reason across multiple, distributed graphs and handle multi-source data while preserving privacy. To overcome these challenges, the authors propose FedNGDBs, a novel framework that integrates federated learning to enable privacy-preserving, cross-graph reasoning. FedNGDBs enhance relational understanding across diverse data sources without compromising data security. Empirical evaluations on three real-world datasets demonstrate that FedNGDBs effectively retrieve relevant information for cross-graph queries, offering a robust solution for large language models and other applications requiring secure, efficient data retrieval.

**Audience:**

Yes

**Claims And Evidence:**

Yes

**Requested Changes:**

Please refer to the cons

**Strengths And Weaknesses:**

### Pros

- This manuscript extends federated graph embedding systems to complex query answering tasks, which is critical for graph holders’ collaboration.
- Based on three public datasets, this manuscript proposes a benchmark for evaluating the retrieval performance of distributed NGDB systems. The benchmark systematically evaluates the retrieval performance facing cross-graph queries.
- This manuscript proposes FedNGDB, a federated neural graph database system that can retrieve answers of complex queries from distributed NGDBs with privacy preserved.

### Cons

- The experimental results lack comprehensive analysis. For example, Table 3 shows that FedNGDB does not consistently achieve the best performance. Specifically, when GQE is used as the graph query encoding model on the FB15k-237 dataset, FedE outperforms FedNGDB. The authors should provide an explanation for this phenomenon.
- The evaluation is overly simplistic and lacks persuasiveness. It is limited to the FB15k-237, FB15k, and NELL995 datasets, all of which are similar in scale regarding the number of training, validation, and test triples. As a result, the experiments do not convincingly demonstrate the effectiveness of the proposed method on larger knowledge graphs.
- The evaluation is limited to scenarios involving only three or five clients in the federated learning system, without any justification for selecting such a small number. The authors should clarify why configurations with more clients, such as ten, were not considered.
- While it is commendable that the authors acknowledge privacy concerns in federated learning, they omit any discussion on backdoor attacks[1, 2, 3], which represent a significant security threat in such systems.
- The authors introduce a parameter-sharing mechanism within the proposed federated learning framework; however, they do not present any experimental results to assess the time overhead associated with this process.

There are some missing references:

[1] Liang et al. A Survey of Knowledge Graph Reasoning on Graph Types: Static, Dynamic, and Multi-Modal. TPAMI 2024.

[2] Yu et al. G${}^2$uardFL: Safeguarding Federated Learning Against Backdoor Attacks through Attributed Client Graph Clustering. arXiv 2023.

[3] Yu et al. GZOO: Black-box Node Injection Attack on Graph Neural Networks via Zeroth-order Optimization. IEEE TKDE 2024.

---

> ### Author Response · Authors · 2025-04-22
> **The reply to Reviewer RKTW**
>
> Thanks for your reviews, and we will respond to your concerns below:
>
> - The experimental results lack a comprehensive analysis.
>
> We want to emphasize that the performance does not consistently outperform the other method in the experiments for several reasons. The reason is that FedE and FedR are designed for federated knowledge graph embedding tasks and are not suitable for distributed complex query answering tasks; they both have their drawbacks. For example, FedE has to share model parameters with the server, which is sensitive. FedR cannot support cross-graph queries. Our method, FedNGDB can provide stronger protection to the whole system while maintaining competitive performance with other methods, such as FedE.
>
>
> - The evaluation is overly simplistic and lacks persuasiveness.
>
>
> We admit that the three datasets used are similar in size. However, these three datasets are the most commonly used datasets in central complex query answering tasks, for example, in [1, 2]. Therefore, we follow the work in central complex query answering and extend the model to a distributed manner, and evaluate the performance on these commonly used datasets so that we can have a better understanding of the retrieval performance, as we can compare the results with the reported numbers from previous works.
>
> [1] Rethinking Complex Queries on Knowledge Graphs with Neural Link Predictors. ICLR 2024.
>
> [2] Is Complex Query Answering Really Complex? Arxiv 2024
>
>
> - The evaluation is limited to scenarios involving only three or five clients.
>
>
> In Appendix C.1, we split the graph-structured data into 10 subgraphs and evaluate the retrieval performance of FedNGDB using GQE as the base model. FedNGDB can still improve the retrieval performance compared to local training when there are more clients participating in the distributed system, demonstrating the effectiveness of FedNGDB.
>
>
> - The discussion of backdoor attacks.
>
> Thank you for highlighting the importance of addressing backdoor attacks in federated learning systems. We acknowledge that backdoor attacks pose a significant security threat. FedNGDB employs privacy-preserving techniques like secret aggregation and differential privacy to enhance security. To address backdoor attacks, we will include a discussion in the revised manuscript, highlighting their significance as a security threat in federated neural graph database systems and proposing potential mitigation strategies as future research directions. We appreciate this suggestion, which will enrich the paper’s security discussion. Besides, we will include the missing references in the revised version.
>
>
> - The time overhead of the parameter-sharing process.
>
> Thank you for your comment regarding the parameter-sharing mechanism in our federated learning framework. We clarify that the parameter-sharing process, described in Appendix A, occurs only during the training initialization phase and constitutes a very small fraction of the overall training time.

---

### Review · Reviewer_x3N4 · 2025-04-18

**Summary Of Contributions:**

This paper proposes FedNGDB for answering complex queries across distributed graph databases while preserving data privacy.
FedNGDB enables collaborative reasoning over multi NGDBs using federated learning and secure computation.
This paper also introduce a three benchmarks to evaluate the performance of complex queries that require information between multi graph databases, claiming that FedNGDB is effective on these benchmarks.

**Audience:**

Yes

**Claims And Evidence:**

Yes

**Requested Changes:**

Please see the weakness section.

**Strengths And Weaknesses:**

**Strengths:**
- The paper is well-written, and the figures are clear and enhance understanding.
- The experimental setup and results are described with clarity and sufficient detail.
- A diverse set of experiments is conducted, enabling meaningful comparisons with several existing methods.

**Weaknesses:**
- Some claims in the manuscript appear overly strong. I would recommend moderating the language used to describe the results. For example, the manuscript states that FedNGDB can effectively retrieve solutions for all benchmarks in Table 3. However, it is not evident from Table 3 that FedNGDB consistently outperforms the other methods. Similarly, the comparisons shown in Figures 4(a) and 4(b) do not clearly demonstrate superior performance of FedNGDB.
- The only difference between Table 3 and Table 4 is the number of clients, so it should be possible to evaluate FedE and FedR in Table 4 as well. However, the results for these methods are missing. Is there a reason why they were not included? It would be important to assess whether the performance of FedNGDB improves with an increasing number of clients, especially in comparison to existing methods.

---

> ### Author Response · Authors · 2025-04-22
> **The reply to Reviewer x3N4**
>
> Thanks for your reviews, and we will respond to your concerns below:
>
> - Some claims appear overly strong.
>
> Thanks for pointing out the weakness. We will double-check all the claims in the manuscript and modify the inappropriate claims. However, we want to emphasize that the performance does not consistently outperform the other method in the experiments for several reasons. The reason is that other methods all have their drawbacks. For example, FedE has to share model parameters with the server, which is sensitive. FedR cannot support cross-graph queries. Central is not a distributed learning method. To the best of our knowledge, there are no suitable methods for the complex queries from distributed graph databases. Our method, FedNGDB can provide stronger protection to the whole system while maintaining competitive performance with other methods, such as FedE.
>
> Besides, for Figure 4, we compare the retrieval performance with the base model trained in central or in distribution without collaboration to further understand the performance of our proposed FedNGDB. For example, we compare the performance of each query type to see the retrieval performance on different types of queries. We compare the participants’ performance to see whether attending the training process will influence the original query retrieval.
>
>
> - The incomplete Table 4.
>
> We compare FedNGDB to central training and distributed training without collaboration, because theoretically, they represent the ideal upper bound and lower bound of the distributed complex query answering performance. The reason why we do not include the comparison to FedE and FedR is that these two methods are not suitable for this task, as I mentioned in the last reply. FedE has to share model parameters with the server, which is sensitive. FedR cannot support cross-graph queries. Therefore, we choose to compare our method once to these two methods to show how the related federated knowledge graph embedding methods perform in this task, although they both have drawbacks. We will conclude the comparison in the appendix in the revision.

---

### Review · Reviewer_kdka · 2025-05-04

**Summary Of Contributions:**

This paper introduces **a novel task** setting focused on answering queries from distributed knowledge graphs. This task is particularly relevant given the rapid growth of real-world graph systems that involve multi-source data and the necessity for cross-graph queries while ensuring privacy through federated learning.

To address this task, the authors extend federated learning techniques, enabling privacy-preserving reasoning across multi-source graph data. Additionally, they propose **a new benchmark** that includes both in-graph and cross-graph queries, effectively addressing the current evaluation gap for such tasks.

**Audience:**

Yes

**Claims And Evidence:**

No

**Requested Changes:**

Please refer to weaknesses

**Strengths And Weaknesses:**

## **Strengths**

* The proposed task is significant and requires further research attention.
* The overall framework is intuitive and achieves promising results.

---

## **Weaknesses**

* **Methodology**

  The methodology section is difficult to follow and lacks essential details. My concerns are as follows:
   * What distinguishes the proposed FedNGDB from existing federated learning methods applied to graph learning or graph database systems? Section 4.1.2 appears to be a simple extension.
   * How are cross-graph and in-graph queries classified? It seems labels are assigned during the query process, but an automatic discrimination between the two types of queries is crucial.
  * The process by which the server decomposes queries into atomic expressions is not adequately discussed. If expressed in natural language, this process resembles task planning in LLM-based agents, yet its concrete execution remains unclear.
   * (Minor point) Figure 2 could be improved; for different clients, the graph data should vary.

* **Experiments**
   * **Benchmark Creation**: Creating a benchmark for cross-graph queries is a significant contribution. However, technical details need justification, such as the choice of 3 and 5 clients, whether the query structures represent all real-world scenarios, and how data quality is controlled or assessed.
   * **Performance**: According to Table 3, FedNGDB does not consistently achieve satisfactory performance and often underperforms compared to baselines like FedE and FedR. Additionally, the omission of these baselines in Table 4 raises questions.
   * For Figure 4, since its purpose is comparison, it would be beneficial to combine (a) and (b) with (c) and (d) into two separate figures. The current separate presentation only allows observation of individual methods' trends, hindering direct comparison and analysis between the two methods.

---

> ### Author Response · Authors · 2025-05-08
> **The reply to reviewer kdka**
>
> Thanks for your reviews, and we will respond to your concerns below:
>
>
> - Further explanation of FedNGDB
>
> For existing federated learning methods, they have to share some sensitive information to conduct reasoning over multiple graphs. To solve the problem, we design a way to process coming queries recursively to prevent the sensitive information leakage in the query process. Compared to FedE and FedR, our proposed FedNGDB can solve the mentioned drawbacks and retrieve similar or better performance.
>
> - The classification of cross-graph and in-graph queries.
>
> In the dataset construction stage, we split the original graph to several parts to simulate the federated setting. In this stage, we sample queries and classify those queries to in-graph and cross-graph queries according to the definition. As we only need to evaluate the performance on these two types of queries.
>
> In the testing stage, we do not classify the in-graph and cross-graph queries. We treat all the queries same: process the queries in all the participants and compute the query encoding and answer score on all the sub-graphs.
>
> - The process of query decomposition
>
> The coming queries are in logical expressions. We decompose the complex queries to atomic queries and then process them using logical operators. For example, we have a query “A and B”, we can first find the query answers of A and B, and apply intersection operator of two answers sets to get the final answers.
>
> - Figure 2
>
> Thanks for pointing out that, we will improve the figure in the revision.
>
>
> - Benchmark creation
>
> We will improve the benchmark creation details justification in the revision. For the questions mentioned by reviewers. Here is our brief reply: For the client number, we follow the common evaluation of federated knowledge graph embeddings papers. For the query structures, we sample the most common query structures as our evaluation. The data quality is evaluated by the existing central neural graph databases
>
> - Performance
>
>
> We want to emphasize that the performance does not consistently outperform the other method in the experiments for some reasons. The reason is that other methods all have their drawbacks. For example, FedE has to share model parameters with the server, which is sensitive. FedR cannot support cross-graph queries. Central is not a distributed learning method. To the best of our knowledge, there are no suitable methods for the complex queries from distributed graph databases. Our method, FedNGDB can provide stronger protection to the whole system while maintaining competitive performance with other methods, such as FedE.
>
> - Figure 4.
>
> Thanks for pointing out that. We will figure out a better way for comparision.

---

### Decision · Action_Editor_Z8mK · 2025-06-16

**Recommendation:** Accept with minor revision

**Audience:**

Yes

**Audience Explanation:**

The paper addresses a significant problem in the field of machine learning and graph databases, specifically the need for efficient and privacy-preserving data retrieval mechanisms.

**Claims And Evidence:**

Yes

**Claims Explanation:**

1. A detailed description of the limitations of existing neural graph databases (NGDBs) and how FedNGDBs address these limitations.
2. Evaluation on three real-world datasets, demonstrating the effectiveness of FedNGDBs in retrieving relevant information from multi-source graph data while keeping sensitive information secure on local devices.
3. Clear explanations of the methodology and results, which provide convincing evidence for the claims made.